# Amikacin liposome and *Mycobacterium avium* complex: A systematic review

Moein Zangiabadian[1☯], Donya Malekshahian[2☯], Erfan Arabpour[2], Sahel Shafiee Dolat Abadi[2], Fartous Yazarlou[3], Narjess Bostanghadiri[4], Rosella Centis[5], AmirHossein Akbari Aghababa[2], Mohammad Farahbakhsh[6], Mohammad Javad Nasiri[2]*, Giovanni Sotgiu[7]*, Giovanni Battista Migliori[5]*

**1** Endocrinology and Metabolism Research Center, Institute of Basic and Clinical Physiology Sciences, Kerman University of Medical Sciences, Kerman, Iran, **2** Department of Microbiology, School of Medicine, Shahid Beheshti University of Medical Sciences, Tehran, Iran, **3** Department of Pharmacy, Comenius University, Bratislava, Slovakia, **4** Department of Microbiology, School of Medicine, Iran University of Medical Sciences, Tehran, Iran, **5** Servizio di Epidemiologia Clinica delle Malattie Respiratorie, Istituti Clinici Scientifici Maugeri IRCCS, Tradate, Italy, **6** Department of Infectious Diseases, Imam Hossein Educational Hospital, School of Medicine, Shahid Beheshti University of Medical Sciences, Tehran, Iran, **7** Clinical Epidemiology and Medical Statistics Unit, Department of Medicine, Surgery and Pharmacy, University of Sassari, Sassari, Italy

☯ These authors contributed equally to this work.
* mj.nasiri@hotmail.com (MJN); gsotgiu@uniss.it (GS); giovannibattista.migliori@icsmaugeri.it (GBM)

**Data Availability Statement:** All relevant data are within the paper and its Supporting Information files.

## Abstract

### Introduction

The prevalence of *Mycobacterium avium* complex (MAC) is increasing globally. Macrolide-based multidrug regimens have been recommended as the first-line treatment for patients with MAC pulmonary disease. However, developing macrolide resistance was associated with poor treatment outcomes and increased mortality. In 2018, the U.S. Food and Drug Administration approved liposomal amikacin for inhalation (LAI) to treat refractory MAC pulmonary disease. The current systematic review aimed to evaluate LAI's outcomes and adverse events in MAC pulmonary disease.

### Methods

The systematic search was performed in PubMed/Medline, EMBASE, and the Cochrane Controlled Register of Trials (CENTRAL) up to March 8, 2022. The search terms included *Mycobacterium avium* complex, MAC, amikacin, and liposomal amikacin.

### Results

After reviewing 1284 records, four papers met the inclusion criteria, including three clinical trials and one prospective cohort study. These studies showed that adding LAI to guideline-based therapies can increase sputum culture conversion rate and achieve early sustained (negative sputum culture results for 12 months with treatment) and durable (negative sputum culture results for three months after treatment) negative sputum culture. In addition, extended LAI use was a potential benefit in patients considered refractory to initial

**Funding:** This study received financial support from the Research Department of the School of Medicine, Shahid Beheshti University of Medical Sciences, Tehran, Iran (Grant number: 32969). The funders had no role in study design, data collection and analysis, decision to publish, or preparation of the manuscript.

**Competing interests:** The authors have declared that no competing interests exist.

treatment. The most prevalent treatment-emergent adverse events (TEAE) reported in the LAI group were the respiratory TEAE.

## Conclusions

LAI could increase the sputum culture conversion rate and achieve early sustainable, durable negative sputum culture. However, additional large-scale research is required to confirm the results.

## Introduction

Nontuberculous mycobacteria are ubiquitous microorganisms that can cause human infections, particularly affecting the lungs [1]. NTM pulmonary disease is becoming more prevalent worldwide, especially among the elderly and immunocompromised people [2, 3]. Symptoms of pulmonary NTM are non-specific and severity is dependent on presence of baseline lung comorbidities such as chronic obstructive pulmonary disease (COPD), bronchiectasis, and cystic fibrosis. The majority (80%) of pulmonary NTM infections are caused by *Mycobacterium avium* complex (MAC) [3]. The prevalence of MAC infection is increasing globally, notably in East Asia and North America, and the frequencies are much higher in individuals aged ≥65 years [4, 5]. Treatment of NTM lung disease is challenging and expensive and could impair the patient's quality of life [6]. The American Thoracic Society/Infectious Diseases Society of America guidelines recommend guideline-based therapies (GBT) containing rifamycin and ethambutol with a macrolide for at least 12 months after culture conversion [1].

The occurrence of macrolide resistance increases the risk of poor treatment outcomes [7]. According to a recent meta-analysis, patients with macrolide-resistant MAC pulmonary disease had an overall sputum culture conversion rate of 21%, and the one-year all-cause mortality rate was 10% [8]. In 2018, the U.S. Food and Drug Administration (FDA) approved liposomal amikacin for inhalation (LAI) in an attempt to achieve a sufficient concentration in the lungs while decreasing the risk of systemic adverse events in adult patients who have not achieved negative sputum cultures despite ≥ six consecutive months of GBT therapy [9]. LAI was developed as a nebulizer to bring high concentrations of amikacin sulfate to the site of infection while limiting systemic exposure [10]. LAI achieves the maximum cell concentration in 2 to 4 hours after administration. After 24 hours, individuals treated with LAI had a 4-fold increase in intracellular amikacin compared to those treated with the same amount of free amikacin [11]. The efficacy and safety of LAI in the treatment of MAC lung disease were evaluated in a number of clinical trials. However, a comprehensive analysis has not yet been performed. Thus, this systematic review aimed to assess the outcomes and safety of LAI in MAC patients.

## Methods

The Preferred Reporting Items for Systematic Reviews and Meta-Analyses statement was used to conduct and report this study [12].

### Search strategy

A search for relevant scientific evidence was conducted using PubMed/MEDLINE, EMBASE, and the Cochrane Library. All clinical studies published up to September 15, 2022, and

reported on the efficacy and safety of ALIS-containing regimens in MAC-infected patients, were retrieved. The search terms were: (((*Mycobacterium avium* Complex [MeSH Terms]) OR (*Mycobacterium avium* [Title/Abstract])) OR (*M. avium* [Title/Abstract])) AND ((Amikacin [MeSH Terms]) OR (Amikacin [Title/Abstract])) OR (liposomal amikacin [Title/Abstract]))) (S1 Appendix, S1 Checklist). Only English-language clinical trials and observational studies were chosen. Further research was found by manually searching the reference lists from selected publications and relevant review articles.

## Study selection

The studies found through database searching were merged, and the duplicates were removed by EndNote X8 (Thomson Reuters, Toronto, ON, Canada). Two reviewers (MZ and EA) independently reviewed them by title/abstract and full text to eliminate the irrelevant to the study's goals. All disagreements were resolved by the lead investigator (MJN). Studies that met the following criteria were included: (1) patients with MAC infection diagnosis, (2) patients receiving liposomal amikacin; (3) treatment outcomes (sputum culture conversion); and (4) safety and adverse effects. The non-liposomal form of amikacin or non-MAC tuberculosis studies, conference abstracts, editorials, reviews, study protocols, and molecular or experimental studies on animal models were excluded.

## Data extraction

Two reviewers (MZ and EA) extracted data from all eligible articles, and any discrepancies were settled by consensus. The following data were extracted: first author's name; year of publication; type of study; the number of patients with MAC infection; patient age, gender, and pulmonary comorbidities; treatment protocols (treatment regimens and duration of treatment); treatment outcomes; definition of treatment success and failure and type of adverse events.

## Quality assessment

Two reviewers (MZ and EA) evaluated the quality of the included records by two different assessment techniques: The Newcastle-Ottawa Scale (NOS) for observational studies and the Cochrane tool for experimental studies [13, 14]. The NOS scale assesses the risk of bias in observational studies across three domains: (1) participant selection, (2) comparability, and (3) outcomes. A study can receive a maximum of one point for each numbered item in the selection and outcome categories and two points for comparability. Low, moderate, and high-quality studies received scores between 0 to 3, 4 to 6, and 6 to 9, respectively.

The Cochrane tool is based on random sequence generation, concealment of allocation to conditions, blinding of participants, personnel and outcome assessors, completeness of outcome data, and other sources of bias. Each study was classified as having a low or high risk of bias when there was no or considerable concern about bias. Besides, unclear risk of bias was defined when information is missing.

## Results

A total of 1,284 records were identified through the initial search (Fig 1); four publications were chosen after removing duplicates and full-text reviews. There were three clinical trials [15–17] and one prospective cohort study [18] among the included articles (Table 1). The total population of the articles was 593. The mean (SD) age of participants was 64.6 (0.28) years [15, 17, 18]. Sputum culture was the most common diagnostic test for MAC in all studies. In the

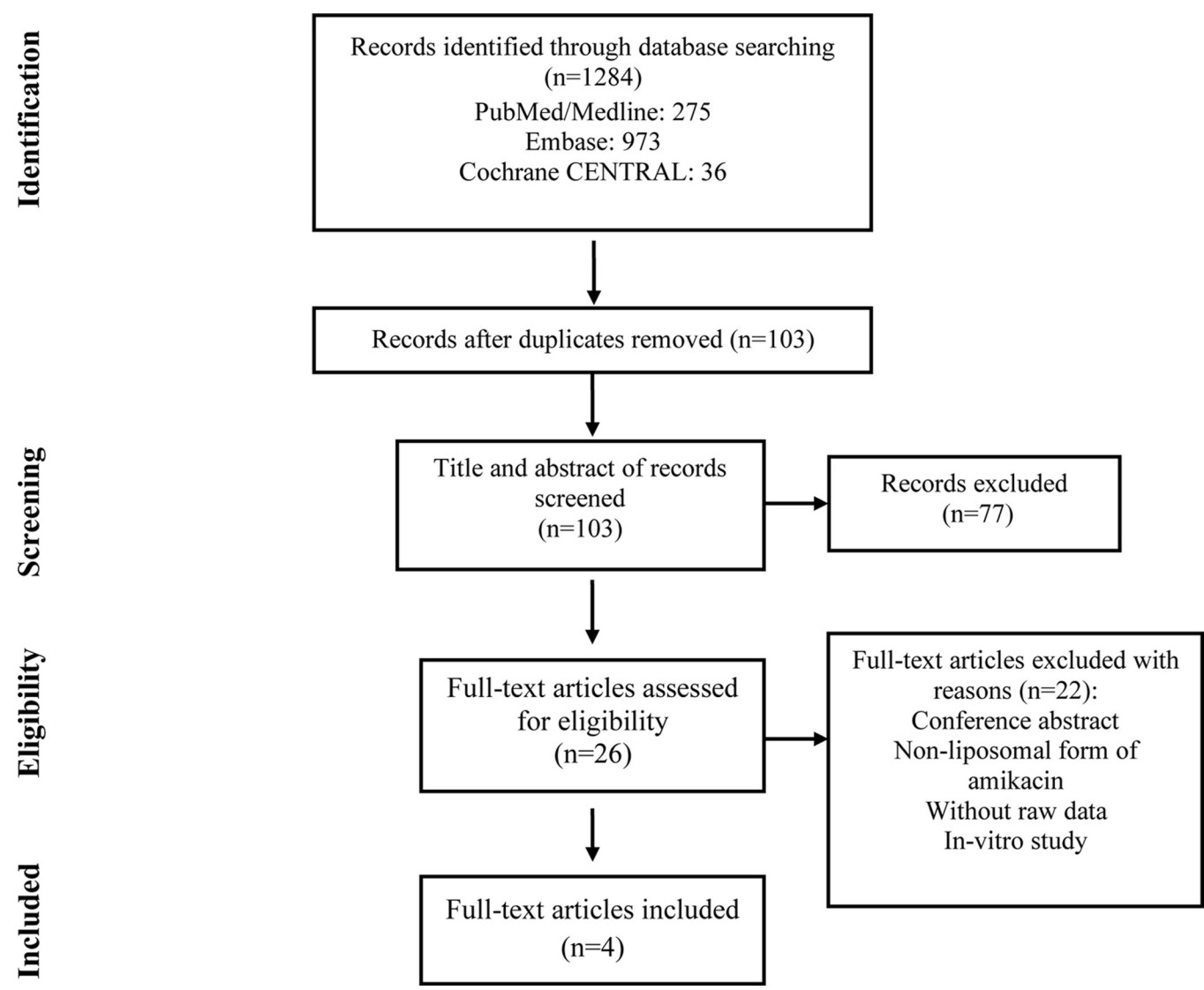

**Fig 1. Study selection flow chart for the studies in the systematic review.**

**Table 1. Included studies.**

| Author | Year | Type of study | No. of participants (M/F %) | Mean age (years) | Pulmonary comorbidities: % |
|---|---|---|---|---|---|
| Griffith et al. [15] | 2018 | Randomized clinical trials | 336 (30.6/69.4) | 64.7 | CF: 0 |
| | | | | | Bronchiectasis only: 210/336 (62.5) |
| | | | | | COPD only: 48/336 (14.3) |
| | | | | | COPD and bronchiectasis: 40/336 (11.9%) |
| Olivier et al. [16] | 2016 | Randomized clinical trials | 19 (NR) | NR | CF: 0 |
| Winthrop et al. [17] | 2021 | Randomized clinical trials | 163 (35.6/64.4) | 64.8 | CF: 0 |
| | | | | | Bronchiectasis only: 116/163 (71.1) |
| | | | | | COPD only: 42/163 (25.7) |
| Griffith et al. [18] | 2021 | Prospective cohort | 75 (18.6/81.4) | 63.9 | CF: 0 |

Abbreviations: NR: NOT REPORTED, CF: cystic fibrosis, COPD: chronic obstructive pulmonary disease

**Table 2. Intervention characteristics.**

| Author | Length of treatment | Treatment regimen | Other drugs included in regimen |
|---|---|---|---|
| Griffith et al. [15] | 12 to 16 months | 590 mg LAI once daily | GBT |
| Olivier et al. [16] | 3 months | 590 mg LAI once daily | GBT |
| Winthrop et al. [17] | 12 months | 590 mg LAI once daily | GBT |

LAI: Liposomal amikacin for inhalation, GBT: guideline-based therapies

included studies, the treatment duration ranged from 84 days to 16 months, and 590 mg of liposomal amikacin was administered once daily along with GBT (Table 2).

## Quality of included studies

Based on the Newcastle-Ottawa Scale, which was used to evaluate the quality of the observational study [18], the NOS score was 9, indicating high methodological quality and a low risk of bias in the included studies (Table 3). Only one experimental study [17] had a high risk of bias for random sequence generation, allocation concealment, blinding of participants and personnel, and blinding of outcomes (Table 4).

## Efficacy of liposomal amikacin on MAC

In Olivier et al. [16] study, the treatment success at day 84 was 91.6% (11/12) and 42.8% (3/7) for LAI and GBT, respectively. At the 12-month safety follow-up, the cultures of 64% (9/14) of patients with *M. avium* complex infection. Which were converted after initiating LAI, had been sustained negative after LAI discontinuation, two patients had positive sputum cultures, and three withheld consent to participate in the 12-month follow-up phase. According to the first study published by Griffith et al. [15] in 2018 (CONVERT study), 65 patients (29.0%) with LAI+ GBT and ten patients (8.9%) with GBT alone had culture conversion (OR, 4.22; 95% CI [2.08, 8.57]; P <0.001). Patients in the LAI+GBT group were more likely to achieve culture conversion compared to the GBT alone group (hazard ratio, 3.90; 95% CI, [2.00, 7.60]). A higher proportion of patients left the study in the LAI+GBT arm (19.6%) because of patient withdrawal (8.5%), adverse events (3.6%), and death (3.1%). In 2021, two studies continued the CONVERT study in different ways. In the INS 312 study, Winthrop et al. [17] chose patients who had a positive culture at the end of the CONVERT study from both prior LAI and LAI-naive cohorts and demonstrated that 26.7% and 33.3% of the LAI-naive cohort get culture conversion by month 6 and 12, respectively. However, similar proportions of patients with refractory disease initiating LAI achieved culture conversion by month 6 in the INS 312 (26.7%) and CONVERT (29%) studies. In 2021, Griffith et al. [17] selected patients with negative culture in month 6 of the CONVERT study to evaluate the culture conversion rate after 12

**Table 3. Quality assessment of the observational studies by The NOS tool.**

| Study | Selection | | | | Comparability | | Outcome | | | |
|---|---|---|---|---|---|---|---|---|---|---|
| | Representativeness of the Exposed cohort | Selection of non-exposed cohort | Ascertainment of exposure | Outcome of interest not present at the start of the study | Adjust for the most important risk factors | Adjust for other risk factors | Assessment of outcome | Adequate length of Follow-up for outcomes to occur | Adequacy of follow-up of cohort | Total quality score |
| Griffith et al. [18] | 1 | 1 | 1 | 1 | 1 | 1 | 1 | 1 | 1 | 9 |

**Table 4. Quality assessment of the experimental studies by the Cochrane collaboration tool.**

| Study | Random sequence generation | Allocation concealment | Blinding of participants and personnel | Blinding of outcome assessment | Incomplete outcome data | Selective reporting | Other bias |
|---|---|---|---|---|---|---|---|
| Griffith et al. [15] | Low risk | Low risk | Low risk | Low risk | Low risk | Low risk | Low risk |
| Olivier et al. [16] | Low risk | Low risk | Low risk | Low risk | Low risk | Low risk | Low risk |
| Winthrop et al. [17] | High risk | High risk | High risk | High risk | Low risk | Low risk | Low risk |

months. 63.1% of patients in the LAI+GBT arm who had gotten culture conversion had a sustained response after 12 months of post-conversion treatment. However, only 30.0% of the GBT-alone arm experienced that. 55.4% of conversions in the LAI+GBT arm had a durable response after three months of treatment, while none did in the GBT group. At month 12 of treatment, 46.2% of the LAI+GBT arm's converts remained culture-negative. Table 5 displays the definitions of treatment success, failure, and outcomes for each group.

## Adverse events

No new safety signals related to 12 months of post-conversion treatment were recorded in the observational study [18]. In the study by Griffith et al. [15] in 2018, treatment-emergent adverse events (TEAE) were reported in 98.2% and 91.1% of patients in the LAI+GBT and GBT-alone groups, respectively. Most TEAEs were moderate in the LAI+GBT group and mild in the GBT-alone group. TEAEs included dysphonia, cough, hemoptysis, dyspnea, fatigue, diarrhea, nausea, and oropharyngeal pain. They were more frequent in the LAI+GBT group, except for hemoptysis, which was equally prevalent in both groups. The most common one was respiratory events (87.4% in the LAI+GBT group and 50.0% in GBT-alone groups), which were mild to moderate. Most TEAEs were initially reported in the first month of treatment,

**Table 5. Treatment success, failure and outcomes of studies.**

| Author | Treatment success | Treatment failure | Outcomes | | |
|---|---|---|---|---|---|
| | | | Success (%) | Failure (%) | Death (%) |
| Griffith et al. [15] (LAI+GBT) | Culture conversion (three consecutive negative sputum cultures monthly by month 6) | NR | 65/224 (29) | NR | 7/224 (3.1) |
| Griffith et al. [15] (GBT) | Culture conversion (three consecutive negative sputum cultures monthly by month 6) | NR | 10/112 (8.9) | NR | 4/112 (3.5) |
| Olivier et al. [16] (LAI) | Culture conversion by month 3 | Complete study without culture conversion by month 3 | 11/12 (91.6) | 1/12 (8.4) | 0/12 (0) |
| Olivier et al. [16] (GBT) | Culture conversion by month 3 | Complete study without culture conversion by month 3 | 3/7 (42.8) | 4/7 (57.2) | 0/7 (0) |
| Winthrop et al. [17] (LAI) | Culture conversion by month 12 | Complete study without culture conversion by month 12 | 10/73 (13.7) | 39/73 (53.4) | 2/73 (2.7) |
| Winthrop et al. [17] (LAI) | Culture conversion by month 12 | Complete study without culture conversion by month 12 | 30/90 (33.3) | 28/90 (31.1) | 1/90 (1.1) |
| Griffith et al. [18] (LAI+GBT) | Culture conversion by month 12 | non-sustained culture conversion by month 12 | 41/65 (63.1) | 24/65 (36.9) | 0 (0) |
| Griffith et al. [18] (GBT) | Culture conversion by month 12 | non-sustained culture conversion by month 12 | 3/10 (30) | 7/10 (70) | 0 (0) |

NR: not reported, GBT: guideline-based therapies, LAI: Liposomal Amikacin for inhalation

**Table 6. Adverse effects in experimental studies.**

| Author | No. of patients | Renal failure/ Increased creatinine | Ototoxicity/ Hearing loss | Hematological disorder | Gastrointestinal symptoms | Peripheral neuropathy | Arthralgia | Psychiatric disorder | Dermatologic symptoms | Death |
|---|---|---|---|---|---|---|---|---|---|---|
| Griffith et al. [15] (LAI +GBT/ GBT alone) | (224/ 112) | NR | 31/10 14%/9% | 10/2 4.5%/2% | 87/24 39%/21.5% | NR | 14/3 6%/3% | 21/5 10%/4.5% | 42/13 19%/12% | 7/4 3%/ 3.5% |
| Winthrop et al. [17] (LAI) | 73/90 | 2/2 3%/2% | 7/7 9.5%/8% | NR | 0/9 0%/10% | 1/1 1.5%/1% | NR | NR | NR | 2/1 3%/ 1% |
| Olivier et al. [16] (LAI/ GBT) | 12/7 | NR | NR | NR | NR | NR | NR | NR | NR | 0/0 |

NR: not reported, GBT: guideline-based therapies, LAI: Liposomal Amikacin for inhalation

and the severe TEAEs occurred in 20.2% and 17.9% of the patients in the LAI+GBT and GBT-alone groups, respectively. According to Winthrop et al. [17] 46.6% of the prior-LAI cohort and 83.3% of the LAI-naive cohort experienced respiratory TEAEs, of which 27.4% and 35.6% of them were severe, respectively. The safety and tolerability of the LAI+GBT were the same in the INS-312 extension study [17] and the CONVERT study. Also, no new safety signs were detected with up to 20 months of LAI exposure; respiratory adverse events were common after LAI exposure (Table 6).

## Discussion

The incidence and prevalence of MAC infection are growing globally [1]. 30–40% of patients with MAC infection fail to achieve sustained culture conversion [19, 20]. According to The American Thoracic Society and the Infectious Diseases Society of America treatment guidelines, individuals with advanced or previously treated MAC lung disease should receive intravenous aminoglycosides (streptomycin or amikacin) [1, 21]. However, the administration of intravascular aminoglycoside is restricted by the risk of ototoxicity and renal toxicity [1]. According to the systematic review by Raaijmakers et al., 7% of patients stopped intravascular amikacin use because of the adverse effects (hearing loss = 7%, tinnitus = 6%, and nephrotoxicity = 2%) [22]. In 2019, Sarin et al. conducted a systematic review to assess the prevalence of ototoxicity in Indian patients with drug-resistant tuberculosis treated with second-line injectables. The incidence of ototoxicity was 10.12%, regardless of audiometry evaluation. For amikacin (61 patients), the prevalence of sensorineural loss was 21.3% (19.7% high-frequency loss and 1.6% low-frequency loss) [23]. In a retrospective cohort by Aznar et al., ototoxicity was reported in 39% of the patients with NTM pulmonary disease who received intravenous amikacin for a median of 7 months experienced ototoxicity, and it was more common in women and was related to the total dose of amikacin per body weight [24].

The FDA approved LAI to deliver high concentrations of amikacin to the lungs in patients with MAC infection who have not achieved negative sputum culture despite at least six consecutive months of GBT [9, 10].

LAI's in vivo efficacy is the same as the injected free amikacin. Rose et al., demonstrated that LAI is highly effective in intracellular *Mycobacterium avium* subsp. *hominissuis* and *abscessus* [25]. Notably, they proved that LAI was superior to the same concentrations of free amikacin. In another experimental model, the authors revealed that aerosolized amikacin

could reduce M. tuberculosis in the lungs as subcutaneously injected amikacin, with a lower dose and less frequent administration [26]. Likewise, the inhaled antibiotic can be effective and practically feasible in cases of *M. tuberculosis* infection [27, 28]. LAI was generally well tolerated in NMT-infected mice [25].

Furthermore, clinical investigations have shown that LAI is well tolerated and has a long-term effect. Phase 2 clinical trials evaluated its safety and efficacy against *Pseudomonas aeruginosa* and MAC in patients with cystic fibrosis (CF) and non-CF bronchiectasis [29–31]. In CF patients infected by *P. aeruginosa*, the safety and tolerability of LAI were supported by the adverse events (AE), serious AE (SAE) profile, as well as similar discontinuation rate and no differences in the renal and audiology safety profile between the LAI and the placebo groups [29].

Based on the pharmacokinetics and pharmacodynamics of LAI, it can cause significant concentrations in sputum and prolonged lung deposition, but the systemic concentration is limited [29].

According to the initial phase 2 randomized study, patients with MAC lung disease had a greater rate of sputum culture conversion when treated with LAI + GBT compared to the placebo (empty liposomes) + GBT [32].

Our systematic review showed that adding LAI to GBT can increase the sputum culture conversion rate and help achieve early sustainable and durable sputum culture negativity. Furthermore, an extended LAI prescription is a potential benefit in patients resistant to the initial therapy.

The main limitations were the poor availability of articles and the same sample in three of four selected studies.

In conclusion, LAI could increase the sputum culture conversion rate and favor early sustainable and durable sputum culture negativity. However, more large-scale studies are required to verify the results.

## Supporting information

**S1 Checklist.**
(DOCX)

**S1 Appendix.**
(XLSX)

## Author Contributions

**Conceptualization:** Rosella Centis, Mohammad Farahbakhsh, Mohammad Javad Nasiri, Giovanni Sotgiu, Giovanni Battista Migliori.

**Data curation:** Erfan Arabpour, Mohammad Javad Nasiri.

**Formal analysis:** Mohammad Javad Nasiri.

**Investigation:** Donya Malekshahian, Erfan Arabpour, Mohammad Javad Nasiri, Giovanni Battista Migliori.

**Methodology:** Moein Zangiabadian, Narjess Bostanghadiri, Rosella Centis, Mohammad Javad Nasiri, Giovanni Sotgiu, Giovanni Battista Migliori.

**Project administration:** Mohammad Javad Nasiri, Giovanni Battista Migliori.

**Supervision:** Mohammad Javad Nasiri, Giovanni Battista Migliori.

**Validation:** Moein Zangiabadian, Fartous Yazarlou, Mohammad Javad Nasiri.

**Writing – original draft:** Moein Zangiabadian, Erfan Arabpour, Sahel Shafiee Dolat Abadi, AmirHossein Akbari Aghababa, Mohammad Javad Nasiri, Giovanni Sotgiu, Giovanni Battista Migliori.

**Writing – review & editing:** Moein Zangiabadian, Donya Malekshahian, Erfan Arabpour, Mohammad Javad Nasiri, Giovanni Sotgiu, Giovanni Battista Migliori.

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
