## [Decision Letter · Decision Letter 0]

14 Oct 2022

PONE-D-22-17884Amikacin liposome and Mycobacterium avium complex: A systematic reviewPLOS ONE

Dear Dr. Nasiri,

Thank you for submitting your manuscript to PLOS ONE. After careful consideration, we feel that it has merit but does not fully meet PLOS ONE’s publication criteria as it currently stands. Therefore, we invite you to submit a revised version of the manuscript that addresses the points raised during the review process.

We look forward to receiving your revised manuscript.

Kind regards,

Mao-Shui Wang

Academic Editor

PLOS ONE

Journal Requirements:

2. Thank you for submitting the above manuscript to PLOS ONE. During our internal evaluation of the manuscript, we found significant text overlap between your submission and previous work in the introduction, methods and results section of your manuscript. We would like to make you aware that copying extracts from previous publications, especially outside the methods section, word-for-word is unacceptable. In addition, the reproduction of text from published reports has implications for the copyright that may apply to the publications. Please revise the manuscript to rephrase the duplicated text, cite your sources, and provide details as to how the current manuscript advances on previous work. Please note that further consideration is dependent on the submission of a manuscript that addresses these concerns about the overlap in text with published work. We will carefully review your manuscript upon resubmission and further consideration of the manuscript is dependent on the text overlap being addressed in full. Please ensure that your revision is thorough as failure to address the concerns to our satisfaction may result in your submission not being considered further.

This study was supported by the Research Department of the School of Medicine, Shahid Beheshti University of Medical Sciences, Tehran, Iran (Grant number: 32969) and is part of the scientific activities of the GTN (Gloal Tuberculosis Network).

Reviewers' comments:

Reviewer's Responses to Questions

**Comments to the Author**

1. Is the manuscript technically sound, and do the data support the conclusions?

Reviewer #1: Yes

Reviewer #2: Yes

2. Has the statistical analysis been performed appropriately and rigorously? 

Reviewer #1: N/A

Reviewer #2: I Don't Know

3. Have the authors made all data underlying the findings in their manuscript fully available?

Reviewer #1: Yes

Reviewer #2: Yes

4. Is the manuscript presented in an intelligible fashion and written in standard English?

Reviewer #1: Yes

Reviewer #2: No

5. Review Comments to the Author

Reviewer #1: The overall manuscript is well presented and thorough. The main drawback from this study is that it is centred around 4 studies only which makes comparisons to others difficult. My suggestion for the authors is to perhaps revise the manuscript and compare against injectable amikacin in clinical cohorts for M. avium treatment. This would mean that the current study can be used as a summary of clinical studies to make clinically informed decisions on whether LAI is beneficial compared to conventional routes of amikacin delivery.

Reviewer #2: Major comments

The current manuscript was carried out to evaluate the outcomes and adverse events of liposomal amikacin for inhalation in patients with MAC pulmonary disease. The undertaken issue is interesting and worth disseminating. However, several changes are needed before the manuscript is accepted for publication.

Specific comments

Abstract

Line 7-8: “patients who have not responded to conventional treatment...”

Please simplify to “refractory MAC pulmonary disease”

Line 9: “patients with MAC disease”

Please revise to “MAC pulmonary disease”

Line 17: “early sustainable, and durable negative sputum culture”

What is the definition of “sustainable, and durable”? Please clarify it in manuscript.

Introduction

Line 1: I suggest describing the basic information of NTM before mentioning the NTM pulmonary disease.

Line 4: “presence of baseline lung comorbidities”

Please describe the lung comorbidities in this sentence.

Example) bronchiectasis, previous TB, cystic fibrosis

Line 5-6: “The prevalence of MAC is increasing globally,”

Please revise to “MAC infection”.

Discussion

Line 1: “The incidence and prevalence of MAC”

Please revise to “MAC infection”

Line 5-6: “in the regimen for patients with advanced or previously treated disease”

What is the definition of “advanced or previously treated disease”? This sentence is little bit unclear. Please revise it.

Line 7-12

These sentences are repeated from lines 18-27 of the introduction part. Please revise or remove the repeated parts in the manuscript.

Line 19-20: “Likewise, inhaled antibiotic can achieve positive outcomes in cases of M. tuberculosis infection.”

The meaning of this sentence is unclear. Does “positive” mean early negative sputum conversion or improved sputum culture conversion rate? Please clarify it.

Line 23: “Phase 2 clinical described its safety and efficacy…”

Please revise to “Phase 2 clinical trials”

Acknowledgement

Line 3: “Gloal Tuberculosis Network”

Revise to “Global Tuberculosis Network”

6. PLOS authors have the option to publish the peer review history of their article (what does this mean?). If published, this will include your full peer review and any attached files.

Reviewer #1: No

Reviewer #2: No

---

## [Author Response · Author response to Decision Letter 0]

22 Nov 2022

Dear Editor

We thank the editor and the reviewers for their comments on our manuscript. Below is our response to each point raised by the academic editor and reviewers. We hope that we satisfyingly addressed them and that the manuscript will be now suited for publication.

Sincerely,

On behalf of all authors,

Mohammad Javad Nasiri

Response to Editor comments: 

Thank you for considering our manuscript. We thank the editor and reviewers for their thoughtful critique and comments. We have carefully edited the manuscript as requested by you and have provided a point-by-point response below. Please find the revised version included. The revisions are highlighted in yellow in the resubmitted manuscript. We hope this meets the established reputation for the quality of your esteemed journal.

1) Thanks for pointing this out. The manuscript is updated with PLOS ONE's style requirements.

2) Thanks for your fair and constructive comment. The duplicated text is rephrased word by word.

3) Thanks for pointing this out. All your queries about financial disclosure are addressed in cover letter and Funding Statement section of the online submission form

4) Thanks for pointing this out. The acknowledgement is corrected (Page 14, line: 267) and funding information is addressed in cover letter and Funding Statement section of the online submission form.

5) Thanks for pointing this out. The data availability statement and captions for Supporting Information files (search strategy and data extraction form) are added. (Page 14, lines: 260-263)

Response to reviewer 1

We appreciate the time and attention you spent in reviewing our manuscript and your thoughtful critique and comment.

Brief descriptions of some clinical cohorts and systematic reviews about safety and efficacy of injectable amikacin are added to the discussion section of manuscript and highlighted in yellow, for comparing the benefits of LAI against conventional routes of amikacin delivery. (Page 12, lines: 213-224)

Response to reviewer 2

Thank you for your willingness to consider our initial manuscript “Amikacin liposome and Mycobacterium avium complex: A systematic review”. We have carefully considered all comments and revised and improved some parts of the original manuscript as requested. We hope this meets the established reputation for the quality of your esteemed journal. The revisions are highlighted in yellow in the resubmitted manuscript.

1) Thanks for your fair and constructive comment. The simplification has been done. (Page 3, lines: 36 and 37)

2) Thanks for pointing this out. The phrase that you are looking for is added. (Page 3, line: 38)

3) Thanks for your recommendation. The term "sustainable" has been changed to "sustained". Sustained and durable culture conversion is defined by negative sputum culture results for 12 months during treatment and for 3 months after treatment respectively. (Page 3, lines: 44-46)

4) Thanks for your suggestion. The basic information of NTM is added. (Page 4, lines: 55-57)

5) Thanks for your recommendation. The description (chronic obstructive pulmonary disease (COPD), bronchiectasis, and cystic fibrosis) is added to the text. (Page 4, lines: 60 and 61) 

6) Thanks for your suggestion. The phrase has been revised. (Page 4, line: 63)

7) Thanks for pointing this out. The phrase that you are looking for is added. (Page 11, line: 207)

8) Thanks for your suggestion. This sentence has been clarified (advanced or previously treated MAC lung disease). (Page 12, line: 212)

9) Thanks for your recommendation. Some of the mentioned sentences have been deleted and some parts have been revised. (Page 12, Lines: 225-227)

10) Thanks for your fair and constructive comment. To clarify the sentence, it is changed to “Likewise, inhaled antibiotic can be effective and practically feasible in cases of M. tuberculosis infection”. (Page 13, lines: 234-235)

11) Thanks for pointing this out. The phrase that you are looking for is added. (Page 13, line: 238)

12) Thanks for pointing this out. The acknowledgement has been corrected in order to journal requirements. And your revised phrase is added in financial information section in online submission form. (Page 14, line: 267)

---

## [Decision Letter · Decision Letter 1]

13 Dec 2022

Amikacin liposome and Mycobacterium avium complex: A systematic review

PONE-D-22-17884R1

Dear Dr. Nasiri,

We’re pleased to inform you that your manuscript has been judged scientifically suitable for publication and will be formally accepted for publication once it meets all outstanding technical requirements.

Kind regards,

Mao-Shui Wang

Academic Editor

PLOS ONE

Additional Editor Comments (optional):

Reviewers' comments:

Reviewer's Responses to Questions

**Comments to the Author**

1. If the authors have adequately addressed your comments raised in a previous round of review and you feel that this manuscript is now acceptable for publication, you may indicate that here to bypass the “Comments to the Author” section, enter your conflict of interest statement in the “Confidential to Editor” section, and submit your "Accept" recommendation.

Reviewer #2: All comments have been addressed

2. Is the manuscript technically sound, and do the data support the conclusions?

Reviewer #2: Yes

3. Has the statistical analysis been performed appropriately and rigorously? 

Reviewer #2: N/A

4. Have the authors made all data underlying the findings in their manuscript fully available?

Reviewer #2: Yes

5. Is the manuscript presented in an intelligible fashion and written in standard English?

Reviewer #2: Yes

6. Review Comments to the Author

Reviewer #2: All comments raised in a previous review have been addressed, and this manuscript is now acceptable for publication.

7. PLOS authors have the option to publish the peer review history of their article (what does this mean?). If published, this will include your full peer review and any attached files.

Reviewer #2: No

---

## [Editor Report · Acceptance letter]

15 Dec 2022

PONE-D-22-17884R1 

Amikacin liposome and *Mycobacterium avium* complex: A systematic review 

Dear Dr. Nasiri:

I'm pleased to inform you that your manuscript has been deemed suitable for publication in PLOS ONE. Congratulations! Your manuscript is now with our production department. 

Kind regards, 

on behalf of

Dr. Mao-Shui Wang 

Academic Editor

PLOS ONE